# Advances in Promoting the Efficacy of Chimeric Antigen Receptor T Cells in the Treatment of Hepatocellular Carcinoma

**DOI:** 10.3390/cancers14205018

**Published:** 2022-10-13

**Authors:** Jie Shen, Dashuai Yang, Youming Ding

**Affiliations:** Department of Hepatobiliary Surgery, Renmin Hospital of Wuhan University, Wuhan 430060, China

**Keywords:** CAR-T, HCC, treatment, tumor microenvironment (TME), cytokines, PD-1 (programmed cell death protein 1)/PD-L1 (programmed cell death ligand 1)

## Abstract

**Simple Summary:**

Hepatocellular carcinoma (HCC) is one of the most common and deadly cancers worldwide. Its insidious onset, rapid disease progression, and poor prognosis make it difficult to treat. Chimeric antigen receptor (CAR)-T cell therapy, a popular treatment in recent years, has demonstrated unsatisfactory results compared with treatment of patients with lymphoma and hematologic cancers. This review examines and discusses studies focused on enhancing the efficacy of CAR-T cells in the treatment of hepatocellular carcinoma in the past five years and aims to serve as a reference for the study of CAR-T cells and promote the development of CAR-T cells in the clinical treatment of hepatocellular carcinoma.

**Abstract:**

HCC, one of the most common and deadly cancers worldwide, develops from hepatocytes and accounts for more than 90% of primary liver cancers. The current widely used treatment modalities are far from meeting the needs of liver cancer patients. CAR-T cell therapy, which has recently emerged, has shown promising efficacy in lymphoma and hematologic cancers, but there are still many challenges to overcome in its application to the clinical treatment of HCC, including osmotic barriers, the inhibition of hepatocellular carcinoma microenvironment activity, the limited survival and killing ability of CAR-T cells, and inevitable side effects, among others. As a result, a number of studies have begun to address the suboptimal efficacy of CAR-T cells in HCC, and many of these schemes hold good promise. This review focuses on advances in the past five years aimed at promoting the efficacy of CAR-T cell therapy for treatment of HCC.

## 1. Introduction

HCC develops from hepatocytes and accounts for more than 90% of primary liver cancers. According to Global Cancer Statistics 2020, it is the sixth most common cancer and the third most common cause of cancer-related deaths worldwide. In China, it is the fifth most frequent cancer and has the second most frequent cancer death rate [1]. Thus, HCC is one of the most prevalent and deadly cancers in the world. It is insidious in origin, asymptomatic in the early stage, and insensitive to conventional radiotherapy [2]. Surgical resection is the most appropriate method for early diagnosis of HCC, and postoperative recurrence is an adverse and common complication. Liver transplantation is feasible for a few patients who meet the transplantation criteria, but the cost is immense. In recent years, research on immune and targeted drug therapy has made continuous progress, which is gradually bringing great benefits to patients with HCC, especially regarding adoptive cell transfer (ACT) therapy, which has good prospects. CAR-T cell therapy, a typical representative of ACT therapy, has been demonstrated to be effective in killing HCC cells with fewer side effects [3] in many studies of HCC cell lines, mouse models [4,5,6], and patients [NCT03198546, NCT02395250]. However, current studies have shown that the efficacy of CAR-T cells in HCC must be enhanced because it is lacking in comparison with application in patients with hematologic and lymphatic cancers. In this review, we analyze the reasons for the poor efficacy of CAR-T cells in the treatment of HCC in addition to presenting a summary of the studies in the past five years that have focused on enhancing the efficacy of CAR-T cells in the treatment of HCC.

## 2. A Brief Introduction to CAR-T Therapy

CAR-T cell therapy is the most representative cellular immunotherapy and involves genetically modifying immune cells (own or donor lymphocytes) to encode receptors that specifically recognize tumor antigens and also modify other genes, such as those encoding cytokines. The cells then proliferate and produce large numbers of CAR cells, which are finally returned to the patient at the suitable dose [7]. Generally, modified CAR cells consist of three parts, namely, an extracellular domain (ECD), transmembrane domain (TMD), and intracellular domain (ICD). The ECD consists of a signal peptide, a single-chain fragment variant (scFv) with heavy and light chain linkages, and a hinge domain (HD) connecting the structure to the TMD, which recognizes and binds tumor-specific antigens. The TMD anchors the CAR structure to the cell membrane, connects the ICD-containing activation signals, and plays a role in signal transduction. ICD provides proliferative and activating signals, and various combinations of ICDs can be used to enhance the antitumor activity of CAR cells [7,8]. Once the specific antigen is recognized by the ECD, CAR cells are activated, and target cells are killed.

In HCC, antigens such as GPC3, AFP, and MUCIN are often upregulated in addition to some tumor stem cell antigens, which are the targets that CAR-T cells are often designed to attack. However, HCC is a solid tumor, and there are many studies targeting CAR-T to some components outside tumor cells, such as cancer-associated fibroblasts (CAFs) and the extracellular matrix (ECM). In addition, CAR-T cells can also be transduced with functional structures or sequences that allow CAR-T cells to have functions that are absent or weak in ordinary T cells, such as sequences that secrete specific chemokines and cytokines, structural units that counteract hypoxia, etc. In summary, as a special immune cell, CAR cells can improve the therapeutic efficacy of HCC in the following aspects: (1) Transport and penetration, (2) Function and activity of CAR-T cells in tumor microenvironment (TME), (3) CAR Structure, (4) Combinatorial targeted CAR-T cells, (5) Targeting multiple antigens or tumor stem cell antigens, (6) Methods of CAR transfer to T cells, and (7) Other options to enhance efficacy. Table 1 outlines the limitations and their causes and possible solutions in CAR-T cell therapy for HCC.

## 3. Transport and Penetration

### 3.1. Local Administration of CAR Cells to the Tumor

The usual intravenous infusion of CAR cells, which are delivered throughout the body due to blood transport, is not well focused at the tumor site. Hence, some scholars have conducted relevant research to address this. In a study of patients with advanced HCC, tumors completely disappeared after 30 days of the intratumor injection of CAR-T cells [9]. Some clinical trials are also exploring intratumoral injection (NCT03130712, NCT02959151, and NCT02862704) and hepatic arterial infusion (NCT01373047 and NCT0280536) to improve poor transport after intravenous injection, and then in the treatment of HCC [12]. It has been demonstrated in mouse models of atypical teratoid/rhabdoid tumors [13] and patients with metastatic breast cancer that intratumor administration of CAR-T cells has faster kinetics, higher potency, and lower systemic levels of inflammatory cytokines than intravenous administration and is well tolerated by patients [37].

### 3.2. Improve the Penetration of CAR Cells

Many T cells fail to reach cancer cells, possibly because tumor cells have reduced or increased secretion of certain vascular-related factors, such as upregulation of endothelin B receptor expression [38] and downregulation of intercellular cell adhesion molecule-1 (ICAM-1) expression [39]. They also prevent T-cell escape from blood vessels. T cell homing may also be hindered by lack of chemokine receptor expression by the tumor or tumor stroma on the surface of T cells as well as a mismatch between chemokines released by the tumor or tumor stroma and chemokine receptors expressed on T cells. Furthermore, the physical barrier of the tumor stroma, as well as hypoxia-derived tumor vascular disorders, also contributes to the low permeability of T cells [40]. Some scholars have engineered CAR-T cells that express chemokines or chemokine receptors, and others have engineered oncolytic viruses (OVs) that secrete chemokines and are used to pretreat tumors. This is because chemokines are able to form a chemotactic gradient that can induce and guide the transport of cells that express matching chemokine receptors. Disrupting the ECM has also been shown to be a viable option.

Adachi et al. designed 7 × 19 CAR-T cells expressing IL-7 and CCL19 whose division number, absolute cell number, and viability were superior to those of conventional CAR-T cells. In addition, histopathological analyses in a solid tumor model revealed increased infiltration of dendritic cells (DCs) and T cells into tumor tissue [41]. Additional studies confirmed the increased infiltration of 7 × 19 CAR-T cells in a mouse model of HCC and showed promising results in patients with HCC [9]. Several CXCR2 ligands (CXCL1, CXCL2, CXCL5, and CXCL8) have been reported to show relatively high expression levels in human HCC tissues and cell lines. Transduction of CXCR2 into CAR-T cells targeting HCC observably accelerates CAR-T transport and tumor site accumulation in vivo [10]. OVs exhibit tumor selectivity, acceptable immunogenicity, and the ability to deliver targeted transgenes to tumors [42]. Meng et al. constructed an adeno-associated virus 2 (AAV-2) overexpressing CCL19, and the infiltration of glypican 3 (GPC3) CAR-T cells in tumor tissue was significantly increased after intratumor injection of OVs [11].

Changes in ECM composition, abundance, and architecture, and the resulting interactions with cells play a profound role in cell and tissue function. Aberrations in ECM remodeling have noticeable effects on the development and progression of cancer [43]. There are various CAR-T schemes for ECM.

Fibroblast activation protein (FAP), a membrane protease, is highly expressed by CAFs. FAP has an immunosuppressive effect, and its overexpression is associated with poor prognosis of various cancers. Researchers have developed a second-generation CAR-T targeting FAP consisting of scFv FAP with CD8 HD, CD3ζ, and 4-1BB ICD. After treatment with these cells, the mouse tumor model shrank by nearly half, and no distinct toxicity was observed [14]. It has also been reported that heparanase secretion by CAR-T cells improves their ability to degrade ECM, thereby promoting CAR-T cell invasion into tumor tissues [15]. They can therefore be used in conjunction with other types of CAR-T cells.

The chemical methods described above are feasible for destroying the ECM, and physical methods such as photothermal therapy have also been reported to be effective. Mild heating of the tumor triggers physicochemical and physiological changes in the tumor tissue, leading to increased infiltration and accumulation of CAR-T cells. The possible mechanism is that mild heating can partially destroy tumor cells and ECM, thus reducing the density of solid tumors and interstitial fluid pressure and enlarging tumor vasculature, thus being more easily infiltrated by CAR-T cells [44].

## 4. Function and Activity of CAR-T Cells in the TME

Once CAR-T cells have successfully infiltrated inside a tumor, they then face obstacles posed by the TME. The glycolytic metabolism of tumor cells results in an environment that is hypoxic, acidic, high in potassium, low in nutrients, and prone to oxidative stress [45]. Of these, high extracellular potassium impairs the function of T cells, inhibiting their differentiation and proliferation [46], and it also leads to upregulation of programmed cell death ligand 1 (PD-1) [47]. As for PD-1, tumor cells frequently upregulate molecular checkpoint ligands such as PD-L1 and galectin-9, which bind to inhibitory receptors on T cells, resulting in a decline in immune cell function. Furthermore, ECM, tumor-associated macrophages (TAMs), and secreted enzymes all affect the function of CAR-T cells and even their survival [48]. Many scholars have conducted a series of studies aimed at countering the inhibitory effects of the TME.

### 4.1. Against Hypoxia

Hypoxia is a fundamental feature of the TME in which the proliferation and high metabolic demands of tumor cells and inefficient tumor vascular system result in a state of inadequate oxygen supply [49]. It is closely linked with cell proliferation, angiogenesis, metabolism, and tumor immune response, all of which further promote tumor progression [50]. The chimeric-antigen-receptor-tumor-induced-vector (CARTIV) platform, based on promoter-responsive elements (PREs) of gamma interferon (IFN-γ), tumor necrosis factor α (TNF-α), and hypoxia, exhibits synergistic activity in cell lines and effective activation in T cells without loss of safety [20]. Another team reported the feasibility of another hypoxia-inducible transcription amplification system (HiTA-System) against hypoxia, and it does not have any prominent hepatic or systemic toxicity in vivo and exhibits significant antitumor activity. The transactivator of this system is co-regulated by hypoxia response elements (HREs) and an oxygen-dependent degradation domain (ODD), which can specifically bind to its target sequence upstream of genes, subsequently initiating their transcription and translation under hypoxia [21].

### 4.2. Antioxidative Stress

On the other hand, the production of reactive oxygen species (ROS) has been found to increase in various cancers, and it has been demonstrated to activate tumorigenic signals, drive DNA damage and genetic instability, and enhance cell survival and proliferation [51] in addition to elevating levels of ROS, which impair T-cell function. Therefore, ROS consumption is also a good counterstrategy. Ligtenberg et al. constructed CAR-T cells expressing both catalase and antibody with dual-station vector, and these cells demonstrated better performance than ordinary CAR-T cells. Because of the increased levels of intracellular catalase, these cells are in an oxidative state with less ROS accumulation in the basal state and upon activation, maintaining antitumor activity even at high H_2_O_2_ levels [19]. Pretreatment of CAR-T cells with auranofin, a gold (I)-containing phosphine compound, would increase their elimination of tumor cells or autologous tumor spheroids, especially under ROS exposure. It has been verified that the increased ROS resistance is dependent on nuclear factor erythroid2-related factor 2 (Nrf2). Hence, Nrf2 activation in T lymphocytes could be used to improve the efficacy of CAR-T [22].

### 4.3. Diminish the Inhibitory Effect of PD-1/PD-L1

Programmed cell death protein 1 (PD-1) is expressed on activated T cells, and its ligand PD-L1 is a major co-inhibitory checkpoint signal that controls T-cell activities. Various cancer types express high levels of PD-L1 and use PD-L1/PD-1 signaling to evade the immune effects of T cells [52]. Blocking PD-L1/PD-1 has been proven, in several studies, to be a viable option for enhancing CAR-T cell therapy for HCC. In mice experiments (including immunodeficient mice), inhibition of PD-1 expression in GPC3-CAR-T cells enhances their killing effect on tumor cells and dramatically reduces tumor load while protecting themselves from exhaustion in combating native PD-L1-expressing HCC [5]. Pan et al. expressed a fusion protein consisting of the PD-1 ECD and CH3 from IgG4 in GPC3-specific CAR-T cells to block the PD-1/PD-L1 pathway. These CAR-T cells had a much higher proliferation capacity after target cell stimulation than control GPC3-CAR-T cells and showed higher tumor suppression than controls in two HCC tumor xenograft models [53]. It has been proposed that CAR-T cells can be modified to improve antitumor activity by secreting PD-1-blocking scFv, which works in both a paracrine and autocrine manner. This protocol is safe and reliable because secreted scFv remains confined to the tumor, potentially avoiding the toxicity associated with systemic checkpoint inhibition. Of note, the human PD-1-blocking scFvs E23, E26, and E27 used in experiments were isolated from a human scFv phage display library (Eureka Therapeutics) [16]. There is also research evidence that PD-1-deficient CAR-T cells based on lentiviral transduction and CRISPR (clustered regularly interspersed short palindromic repeats)-associated protein (Cas9) ribonucleoprotein particle (RNP)-mediated gene editing is a feasible solution of improving therapeutic efficacy [17]. In addition to the above, by blocking PD-L1/PD-1 recognition, Liu et al. investigated another option to limit the inhibition of signaling even when PD-L1 and PD-1 recognize each other. The team inserted a genetically engineered switch-receptor structure containing the ECD of PD-1 fused to the TMD and ICD of CD28 into CAR-T cells. When the PD1 portion of this switch receptor is in contact with its ligand PD-L1, it delivers an activation signal (via the CD28 ICD) [18], such as increased extracellular signal-regulated protein kinase (ERK) phosphorylation, increased cytokine secretion, increased proliferative capacity, and enhanced expression of the effector molecule granzyme B [54], rather than the suppressive signal normally transduced by PD-1.

### 4.4. Promote the Secretion of Cytokines

Many cytokines regulate the differentiation, activity, and function of T cells, which are closely associated with bodily immune function [55]. Therefore, many studies have focused on cytokines in improving the efficacy of CAR-T cells. Improving the efficacy of CAR-T cells requires promoting the secretion of cytokines to enhance T-cell function.

IL-23 is known to function mainly through autocrine mechanisms to promote T cell proliferation and survival. T cells expressing p40 subunit were shown to produce IL-23 upon activation. Compared with normal CAR-T cells, those with the p40 subunit show improved antitumor capacity, increased granzyme B, and decreased PD-1 expression [23]. Makkouk developed the first allogeneic GPC3-CAR Vδ1 T cells expanded from peripheral blood monocytes that also encode the secreted form of IL-15 (sIL-15). These cells allowed effective control of tumor growth in a murine subcutaneous model with no evidence of graft-versus-host disease (GVHD). Compared with GPC3 CAR Vδ1 T cells lacking sIL-15, they exhibit greater proliferative capacity and enhanced therapeutic efficacy [24]. The research demonstrated that IL-15 can enhance the efficacy of CAR-T cells, and co-expression of IL-15 and IL-21 has been reported to enhance the survival, proliferation, and function of CAR-T cells and improve their antitumor properties. This is partly due to the maintenance of transcription factor-1 (TCF-1) expression by CAR-T cells co-expressing IL-15 and IL-21 as well as a notable reduction in IL-13 production. TCF-1 plays a role in enhancing the expansion and survival of CAR-T cells, whereas IL-13 reduces the antitumor function of T cells and promotes tumor cell proliferation [56,57]. IL-21 can regulate the function of CAR-T cells through other pathways in addition to the methods described above. Wang reported on the novel IL-4 and IL-21 incorporation of inverted cytokine receptor (ICR), which enhances the efficacy of CAR-T cells in the IL-4 tumor environment. The 4/21 ICR was constructed by fusing the ECD of the IL-4 receptor to the TMD and ICD of the IL-21 receptor. Upon IL-4 stimulation, 4/21 ICR activates the signal transducer and activator of the transcription (STAT)3 pathway and upregulates IL-21, eventually promoting Th17-like polarization and tumor-targeted cytotoxicity in CAR-T cells in vitro. In addition, in vivo, 4/21 ICR-CAR-T cells can persist for a long time and eradicate tumors that have established IL-4 expression. Thus, 4/21 ICR is a promising clinical CAR-T cell therapy for IL-4 enriched HCC [58].

### 4.5. Other Solutions to Bypass TME Inhibition

In tumor immunity, regulatory cells (Tregs) participate in the occurrence and development of tumors by inhibiting antitumor immunity. There are several Treg immune suppressive mechanisms: inhibition of costimulatory signals CD80 and CD86 expressed by dendritic cells through cytotoxic T-lymphocyte antigen-4 (CTLA-4); secretion of inhibitory cytokines; regulation of tryptophan and adenosine metabolism; and direct killing of effector T cells [59]. The efficacy of CAR-T cells in HCC is limited, partly due to immunosuppression by Tregs. Although lymphodepletion can often be used to consume Tregs, this method has several severe side effects and a narrow treatment window. Suryadevara et al. reported that the CD28 cytoplasmic tail of the CAR transgene contains a distal PYAP signaling motif that recruits the Src family kinase lymphocyte-specific tyrosine kinase (Lck) to transduce the signal necessary for IL-2 secretion upon CAR activation, which enhances the generation of Tregs. They replaced the PYAP Lck binding motif in the CD28 domain with two amino acids to disable its function, which improved the antitumor efficacy of CAR-T cells.

The transforming growth factor-β (TGF-β) signaling pathway participates in TME regulation through a variety of ways, affecting cell growth, differentiation, wound healing, apoptosis, immunosuppression, etc. In particular, TGF-β is involved in the inhibition of cytotoxic T lymphocytes (CTLs) and upregulation of Tregs [25]. Knockdown of endogenous TGF-β receptor II (TGFBR2) in CAR-T cells by CRISPR/Cas9 can reduce Treg transformation and prevent depletion of CAR-T cells. In their experiment, CAR-T without TGF-β knockdown failed to completely eliminate tumors of 200–300 mm^3^ size at day 49, whereas their designed CAR-T cells had completely cleared tumors at day 35. This means that TGFBR2-edited CAR-T cells have higher tumor elimination efficacy in solid tumor models, both locally and systemic [27]. 

Oncolytic immunotherapy with engineered adenoviruses (OAd) may disrupt the TME by infecting tumor cells and the surrounding matrix to improve CAR-T function. Effectively delivering OAd to solid tumors, however, has proved a challenge. It has been reported in recent years that mesenchymal stromal cells (MSCs) can be used to systemically deliver OAd. MSCs were infected with a binary vector that contains OAd expressing IL-12 and PD-L1 blocker and a helper-dependent Ad (HDAd), thereby allowing the MSCs to express IL-12 and checkpoint PD-L1 blocker. The combination of these MSCs and CAR-T cells improved the infiltration ability and cell viability of CAR T cells [60]. As mentioned above, high extracellular potassium impedes T cell action, and while Ong et al. demonstrated that pharmacological activation of K channels in T cells would reduce immunosuppression within TME and promote T cell-mediated tumor destruction, they identified an activator of K channels, SKA-346, that used alone or with immune checkpoint inhibitors could enhance antitumor function within TME [47]. The effectiveness of CAR-T cells in treating HCC needs to be demonstrated, but it is theoretically feasible.

## 5. CAR Structure

It is not difficult to understand that if CAR-T cells perform satisfactorily, besides the need to get rid of the constraints of the environment, self-optimization is another direction to consider, and this has been confirmed by a number of studies.

### 5.1. Improve HD/TMD

Functional analysis of CAR-T cells has shown that HD affects the transport efficiency of proteins to the cell membrane, whereas TMD regulates the stability of CAR membrane expression. The membrane transport rate of each CAR-T is hypothesized to depend on the folding efficiency of the CAR ECD, which is the efficiency of immunoglobulin domain formation in scFv, whereas the folding efficiency depends on the length of the HD [61]. This has been studied through mutation of the 27th amino acid from cysteine to serine in the CD8α HD away from the cell membrane, resulting in CAR cells with higher killing efficiency, which may be due to formation of cysteine disulfide bonds [62].

### 5.2. Intracellular Domain Optimization

A typical CAR ICD is divided into STD and costimulatory domain (CSD), in which CD28 and/or 4-1BB constitute the CSD, and CD3 is the signal transduction domain, namely, the second- or third-generation CAR structure (Figure 1). Figure 1 depicts a schematic of the basic structure and transgene of CAR cells. This CAR structure is stable but its efficacy in solid tumors needs to be enhanced [63]. CD28 can promote the production of IL-2, IL-6, IL-10, and interleukins in addition to enhanced apoptosis, cell cycle progression, cellular metabolism survival, and differentiation [28]. Wan et al. reported that inducible costimulator (ICOS), a CD28-family costimulatory receptor, is upregulated by T cells after antigen stimulation, then activates phosphoinositide-3 kinase (PI3K) and enhances T-cell receptor (TCR)-triggered calcium mobilization, ultimately improving CAR-T functionality [29]. CAR cells containing 4-1BB as a CSD possess stronger cell activation and higher survival in vivo. The inclusion of 4-1BB CSD promotes differentiation of CAR-T cells toward a central-memory phenotype [64]. On the basis of the second-generation 4-1BB CAR, a molecular transfer switch (T3/28) was added; that is, the ECD of T1M-3 molecule and the ICD of CD28 molecule were fused, shifting the T1M-3 inhibitory signal to an activating signal, which has been shown to improve the ability of CAR-T cells to sustainably kill tumors in vivo [65]. Kagoya’s team developed a novel CAR T cell that activates the Janus kinase (JAK)–STAT pathway. In addition to CD3z and CD28 domains, the ICD of this CAR also contains a truncated cytoplasmic domain of IL-2Rβ and a STAT3-binding YXXQ motif. In leukemia and solid tumor models, it has shown excellent persistence and antitumor effects [66]. In addition to these approaches, CD79A and CD40, which normally interact with B-cell receptor (BCR), can be developed into a complex CSD that cooperates with T-cell signaling to improve the function and persistence of CAR-T cells. Compared with CD28 or 4-1BB, CD79A/CD40 CAR-T cells showed strong cell proliferation and were more capable of inhibiting target cells [67].

## 6. Development of Combinatorial Targeting CAR-T Cells

Tumor specific antigens (Ags) are rare or even absent. The antigens commonly upregulated in HCC, such as AFP, GPC3, etc. are not specific in a strict sense (Table 2). Table 2 shows the current HCC targets and status of research that can be used for CAR-T therapy. When CAR-T cells are designed to target these Ags, they may indiscriminately attack other tissues expressing these Ags [68]. Combined targeting of CAR-T cells, that is, either co-expressing several CARs on a single T-cell (dual CAR-T) or expressing a chimeric receptor with two or more antigen recognition domains (tandem CAR-T), can kill tumor cells that are positive for both target Ags without significant growth inhibition of single- or double-antigen-negative xenografts. According to a report, GPC3/ASGR1 (asialoglycoprotein receptor 1) dual-targeted T cells exerted superior anticancer activity and persistence in HCC xenograft models and had substantially higher cytokine secretion, proliferation, and anti-apoptotic capacity compared with single-targeted T cells [32]. Yang et al. developed tandem CAR-T targeting CD70 and B7-homolog 3 (B7-H3), which was reported to be effective in killing multiple solid tumors, including HCC, reducing the tumor load, and demonstrating superiority to single specific CAR-T cells [33]. Other studies have shown that GPC3/CD3 [69] and GPC3/CD47 [70] bispecific antibodies can also kill HCC cells safely and efficiently, which are superior to monotherapy and the combination of two monoclonal antibodies. Of these, CD47 is a suppressive innate immune checkpoint that interacts with signal regulatory protein alpha (SIRPα) on myeloid cells (especially macrophages) and allows cancer cells to evade immune surveillance [71].

## 7. Targeting Multiple Antigens or Tumor Stem Cell Antigens

Solid tumors are spatially organized heterogeneous ecosystems: at the cellular level, they are tumor and stromal cell populations; at the genetic level, they are derived from the continuous genetic cloning of tumor cells; and at the phenotypic level, they acquire different phenotypes in different microenvironments [80]. Therefore, most tumors exhibit striking physiological and morphological heterogeneity during clinical treatment, even for the same type of cancer, such as HCC [81]. This is one of the main reasons why CAR-T is more effective than solid tumors in the treatment of hematological malignancies, etc. Hematologic and lymphoid cancers have specific tumor antigens, e.g., CD13, CD33 for acute promyelocytic leukemia, CD19 for diffuse large B cell lymphoma, etc. Antigens commonly expressed in hepatocellular carcinoma, such as GPC3 and AFP, vary greatly in the amount and type of expression among individuals with HCC. This challenge can be resolved by using either dual CAR-T or tandem CAR-T to identify multiple antigens [34]. In addition, some antigens associated with tumor stem cells can be targeted because they are expressed in many solid tumors and at different stages of cancer [82]. Targeting multiple antigens has been discussed above. As for antigens associated with tumor stem cells, CD133 is an antigen associated with HCC stem cells. A phase 1 clinical trial involving CD133-targeted CAR-T cells for advanced HCC (NCT02541370) demonstrated its feasibility, manageable toxicity, and efficacy [35]. Phase II of this clinical trial also confirmed the efficacy of CD133 CAR-T cells, although some side effects such as hematologic toxicities were observed, but within a manageable and acceptable range [36].

## 8. Transduction of CARs into T Cells

Most clinical trials use viruses (retroviruses or lentiviruses) to stably transduce CARs [48,83,84]. However, studies have demonstrated that CAR-T is immunogenic, which limits the function and survival of CAR-T cells. In particular, the epitopes encoded by viral vectors may induce cellular immunity [85]. DNA transposon systems, particularly those based on Sleeping Beauty, piggyBac, and Tol2, have emerged as promising alternatives to avoid some of the side effects associated with the use of viral vectors. Trials have shown that they dramatically increase progression-free survival (PFS) and overall survival (OS) in patients with non-Hodgkin’s lymphoma and leukemia and cause less cytotoxicity [30,31]. However, some studies have reported that piggyBac and Sleeping Beauty contain both gene inactivation and activation elements. This mutagenic transposon may greatly increase tumor formation or promote tumorigenesis in mice [86,87]. Whether they are used for insertion of CARs into T cells or cancer treatment, there is an increased probability of inducing cancer, which needs to be further studied.

## 9. Other Options to Enhance Efficacy

Many other studies have taken a unique approach, either developing more efficient recognition modules, using novel materials for CAR-T cells, or combining them with certain anti-liver cancer drugs, or enhancing the secretion of some transcription factors to enhance T-cell function.

Traditional antibodies have two variable domains, termed variable domain of heavy chain (VH) and light chain (VL), which provide stability and determine binding specificity to each other. Xie’s team replaced scFv with variable domain of heavy chain of heavy-chain (VHH) antibody as the recognition module and developed single-domain antibody (nanobody) CAR-T cells based on VHH. Although lacking the VL domain, this type of cell remains highly stable. VHHs are readily displayed as CAR recognition modules and fully reserve specificity of Ag binding. Hence, they have specific cytotoxicity. The experimental results showed that they significantly inhibited tumor growth. The team also further reported that the VHH-based CAR is highly modular and widely applicable to a variety of tumors [88]. Essentially, they have developed a more efficient and convenient recognition structure than the usual scFv.

Tang et al. used protein nanogels (NGs) to “package” a large number of supportive protein drugs onto CAR T cells, which can selectively release this cargo upon T cell activation, thus limiting drug release to sites where Ags are encountered, such as in the TME. It has been confirmed that this method is safe under a wide range of conditions [89]. Intrinsically, this approach is similar to enhancing the secretory function of CAR-T cells, but it is more powerful because of the large and artificially controllable range of drug choices.

Sorafenib is the earliest molecular targeted drug used for systemic antitumor therapy in HCC. Sorafenib is currently an effective first-line treatment in advanced HCC [90]. In mice, subpharmacological doses of sorafenib (pharmacological dose is recommended by the US Food and Drug Administration) enhanced antitumor activity, penetration, and IFN-γ production of GPC3 CAR-T cells in vivo. In addition, GPC3-CAR-T cells can infiltrate mouse tumor tissue and persist as effector memory T cells [6]. Whether other first-line anti-HCC drugs such as atezolizumab, lenvatinib, etc. can enhance the efficacy of CAR-T cells needs to be studied experimentally. Moreover, given that sorafenib facilitates CAR-T infiltration into HCC, it is understandably associated with its own tumor-growth -inhibiting effect.

T-box expressed in T cells (T-bet) was originally described as key transcription factors that define type 1 T-helper cells (Th1). However, a growing body of research suggests that it drives the coordinated production of effector and memory cells in several different lymphocyte lineages and is essential for maintaining immunity [91]. We can design CAR-T cells with high expression of T-bet by gene editing. T-bet (STOP) and T-bet (∆TBOX) can be used to improve the expression of T-bet in CAR-T cells. Of these, T-bet (STOP) was generated by mutating nucleotide 214G→T. T-bet (∆TBOX) was generated by deletion of T-bet nucleotides 403–978 [26]. This approach focuses on maintaining the function and activity of CAR-T cells within the tumor.

Inhibition of glycogen synthase kinase 3 (GSK3) in T cells leads to robust T cell proliferation and memory cell production, partly because blocking GSK3 protects these CAR-T cells from activated T cell death cell and, moreover, the PD-1 level of CAR T cells is decreased, reducing levels of the PD-L1/PD-1 inhibitory signal. It has been verified that SB216763 can be used to inhibit GSK3, and it affects both the α and β subunits of GSK3. In addition, TWS-119 is specific for β subunits and can also be used to block GSK3 [92]. This option mainly improves the proliferation ability of CAR-T cells while blocking the PD-1/PD-L1 pathway. In general, although the methods are not identical, the basic involved principles have been discussed in the above-mentioned content.

## 10. Conclusions

CAR-T cell therapy has shown excellent results in hematological tumors and lymphomas, but in solid tumors such as HCC, the efficacy of this approach is not satisfactory due to barriers in the TME and immune cell penetration. Promising applications have been revealed for the existing research protocols aimed at enhancing the efficacy of CAR-T cells, such as enhancing the infiltration of CAR-T cells and bypassing inhibition of the TME, optimizing the common structure of CARs, developing dual-targeted CAR-T cells, and improving the method of transferring CARs into T cells. The above schemes were discussed in this review. As a new and advanced treatment method, CAR-T cell therapy is expected to be used in the treatment of patients with HCC as soon as feasibly possible, which is good news for patients with HCC.

## Figures and Tables

**Figure 1 cancers-14-05018-f001:**
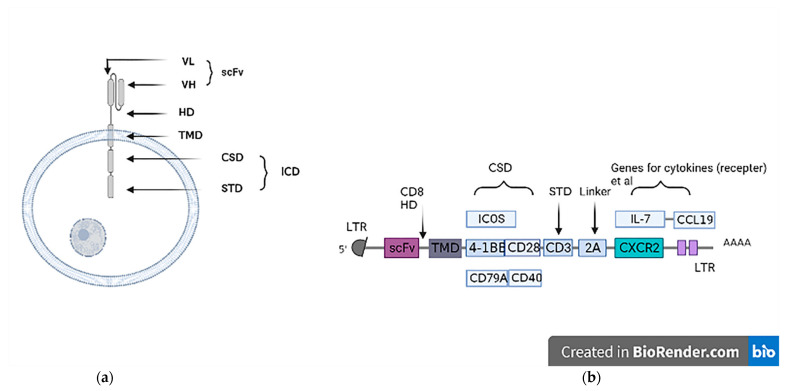
(**a**) Structure of CAR-T cells; (**b**) schematic of the transgene of a third-generation CAR. Long terminal repeat-retrotransposons (LTR), extracellular domain (ECD), transmembrane domain (TMD), intracellular domain (ICD), hinge domain (HD), costimulatory domain (CSD), variable domain of heavy chain (VH), variable domain of light chain (VL), signal transduction domain (STD), and single-chain variable fragments (scFv).

**Table 1 cancers-14-05018-t001:** Limitations of and available strategies for CAR-T cell therapy for hepatocellular carcinoma.

Limitations of CAR-T for HCC	Reasons	Available Strategies
Infiltration obstacles for CAR-T cells	Physical barrier of tumor stroma (extracellular matrix, disturbed vasculature), obstacles to T-cell penetration of blood vessels, lack of chemokine receptors or mismatch with chemokines	Design CAR-T cells expressing chemokines or chemokine receptors [9,10], pretreat with oncolytic virus expressing chemokines [11], local administration (within the tumor, etc.) [12,13], disruption of degraded extracellular matrix [14,15]
Inhibition of CAR T cell function and activity	Tumor microenvironment of HCC (Inhibitory receptor-ligand interaction, acid, hypoxia, oxidative stress, high potassium level, immunosuppression of regulatory T cells and myeloid-derived suppressor cells, negative regulation of cytokines), structural design flaws in CAR itself	Block PD-L1/PD-1 (expresses PD-1 antibody in CAR-T [5,16], modifies the sequence encoding PD-1 [17], develop switch-receptor of PD-1) [18], activate K channels, secrete catalase in CAR-T [19], develop antihypoxia systems [20,21], enhance expression of Nrf2 [22], secrete pro-inflammatory cytokines [23,24], consume regulatory T cells [25]
Viability and persistence of CAR-T cells	Immunogenicity of CAR-T cells themselves, nutritional competition for tumor cells, harsh tumor microenvironment	Increase T-bet expression [26], suppress the function of transforming growth factor-β [27], optimize the intracellular domain [28,29], use the DNA transposon system [30,31]
Antigen escape	Impairment of antigen-presenting ability, lack or downregulation of tumor-specific or associated antigens, lack of costimulatory molecules	Combinatorial targeted CAR-T cells [32,33], two or more CAR-T cells in combination
Tumor heterogeneity	Striking physiological and morphological heterogeneity of HCC	Co-express several CARs on a single T-cell (dual CAR-T), express a chimeric receptor with two or more antigen recognition domains (tandem CAR-T) [34], target tumor stem cell antigens [35,36]

**Table 2 cancers-14-05018-t002:** Alternative targets and research results for liver cancer.

Available Targets for Liver Cancer	Brief Introduction	Clinical Trials	Current Research Results in Liver Cancer	Reference
AFP (alpha-fetoprotein)	A glycoprotein, is usually overexpressed in HCC, germ cell neoplasms, pancreatic cancer, gastrointestinal cancer, or lung cancer	NCT03349255 (phase1) NCT03971747 (phase1) NCT04368182 (phase1)	Sun confirmed that engineered cytotoxic T cells with AFP158-166 specific TCR were effective against liver cancer [72]; 15 patients with liver cancer were enrolled in a phase 1 trial: AFP-derived peptide (AFP357 and AFP403) vaccine can induce the increase of AFP-specific T cells. No severe adverse events related to the drug were observed. 1 patient developed fever, which resolved spontaneously after 1 day, and 1 patient developed gastrointestinal symptoms. Injection-site reactions at the injection site occurred in almost all patients, but they were not severe. After treatment, 1 patient had a complete response, 8 had stable disease [73].	[72,73]
GPC3 (glypican 3)	A 70 kDa heparan sulfate proteoglycan that is undetectable in the liver of healthy adults, but overexpression has been detected in patients with HCC, also expressed in adult tissues ovary, breast, mesothelium, lung and kidney	NCT02395250 (phase1) NCT03130712 (phase1, 2) NCT02715362 (phase 1, 2) NCT03084380 (phase1, 2)	GPC3 has been demonstrated in several trials (liver cancer cell lines, mouse tumor models and liver cancer patients) to be relatively safe and effective in killing tumor cells and reducing tumor size. However, it may cause side effects such as cytokine release syndrome [74], neurotoxicity [75], etc.	[74,75]
NY-ESO-1 (New York esophageal squamous cell carcinoma-1)	Expression is mainly restricted to testicular germ cells and placental trophoblast cells, no or low expression in normal adult somatic cells but ectopic expression in many tumor types	NCT03175705 (phase1) NCT03941626 (phase1, 2)	HLA-A2-restricted NY-ESO-1-specific T cell receptor engineered T cells can kill tumor cells, smaller tumor weight in mice after CAR-T cell treatment compared to control group (*p* = 0.0018) [76]	[76]
MUC1 (mucin-1)	A transmembrane glycoprotein, can expose epitopes that are normally hidden, is widely distributed and exceptionally abundant on the surface of cancer cells (breast cancer, stomach cancer, colon cancer, and liver cancer)	NCT02587689 (phase1, 2) NCT02839954 (phase1, 2)	Anti-MUC1 CAR based on variable domain of heavy chain of heavy chain antibody was effective against MUC1-positive cell lines [77,78]	[77,78]
EpCAM (epithelial cell adhesion molecule)	A 40 kDa type I transmembrane glycoprotein, contributes to a variety of biological processes, including cell adhesion, signaling, migration, and proliferation	NCT03013712 (phase1, 2)	CAR-T cells can be engineered to target EpCAM and show toxicity to antigen-positive tumor cells [79]	[79]

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
