# Peer review of "Advances in Promoting the Efficacy of Chimeric Antigen Receptor T Cells in the Treatment of Hepatocellular Carcinoma"

_cancers, 2022, doi:10.3390/cancers14205018_

Round 1

Reviewer 1 Report

The manuscript presented by Shen Jie et al is of considerable interest and summarizes well the research in cell therapy of HCC. The manuscript could be recommended for publication, but it is necessary to point out some shortcomings. First of all more clinical data should be included and discussed.

section 3
3.1 and 3.2 same caption

section 5
"It is not difficult to understand that if CAR-T cells perform satisfactorily, besides the 257
need to get rid of the constraints of the environment, self-optimization is another direction 258
to consider, and this method has been confirmed by a number of studies. "- consider revising

how does this section relate directly to HCC therapy?

section 6
The results of clinical trials should be covered in more detail, indicating the effectiveness of therapy in the form of numerical indicators.

section 7
The authors should mention that the use of PB for CAR delivery is associated with tumorigenesis.

section 8
"Theoretically, they have specific cytotoxicity" - why theoretically?

Author Response

Point1: section 5
"It is not difficult to understand that if CAR-T cells perform satisfactorily, besides the 257
need to get rid of the constraints of the environment, self-optimization is another direction 258
to consider, and this method has been confirmed by a number of studies. "- consider revising

Response1: It is not hard to understand that if you want CAR-T cells to perform satisfactorily, in addition to freeing them from the constraints of the environment, optimizing their structure is another approach to consider. Importantly, the feasibility of the latter has been proven by many studies.

Point2: how does this section (section 5) relate directly to HCC therapy?

Response2: The structure of CAR-T cells, including extracellular domain (ECD), transmembrane domain (TMD), intracellular domain (ICD), greatly determines their ability to kill tumor cells. Fujiwara et al (doi: 10.3390/cells9051182.) indicated that the hinge domain regulates the CAR signaling threshold and the transmembrane domain regulates the amount of CAR signaling via control of CAR expression level. in other words, HD affected the transport efficiency of proteins to the cell membrane, furthermore, the anti-tumor ability of CAR cells was affected. Qin et al (doi: 10.1186/s13045-017-0437-8.) demonstrated that hinge incorporation mainly promotes CD4+ CAR-T cell expansion during the in vitro culture period and hinge incorporation enhanced the migratory and invasion capabilities of CAR-T cells, which increase the antitumor efficacy of some specific CAR-T cells. Hirobe et al (doi: 10.3390/ijms23074056.) showed that N-linked glycosylation of CAR[V/28/28/3z] promotes stable membrane CAR expression, while having no effect on the expression or CAR-T cell activity of CAR[V/8a/8a/3z], which demonstrates that post-translational modifications of the CAR HD influence CAR-T cell activity. As for intracellular domain, ICD of the third generation CAR-T consists of 4-1BB, CD28 and CD3. CD28 can promote the production of IL-2, IL-6, IL-10 and more interleukins, apoptosis, cell cycle progression, cellular metabolism survival, differentiation, etc. 4-1BB allowed for a more robust cell activation, an increased persistence and a tendency to differentiate into central-memory phenotype. Studies have shown that modifying them improves the above abilities. (doi: 10.1038/s41423-018-0183-z. doi: 10.1136/jitc-2021-003176.). It is also possible to add sequences to make CAR cells more functional through linkers, such as activating JAK-STAT, secreting chemokines, transcription factors (doi: 10.1038/nm.4478. doi: 10.1038/nbt.4086. doi: 10.1016/j.it.2017.02.003.). Overall, the currently used CAR structures can be optimized to show more potent antitumor efficacy. Although many of the results have not been directly confirmed in hepatocellular carcinoma cell lines or models, it has been demonstrated on other cell lines and models that optimal modification to modify their own structure is feasible and effective. Theoretically, the HCC efficacy of modified CAR-T cells will be stronger compared to the normal CAR structure.

Point3: section 8
"Theoretically, they have specific cytotoxicity" - why theoretically?

Response3: Because VHHs fully reserve specificity of Ag binding, in theory, their cytotoxicity is also specific, and subsequent experiments confirmed the results. So we should add that their specific cytotoxicity is proven or replace "theoricaly" with something else.

Other comments  have been responsed in the manuscript.

Regarding English writing, I have applied for English editing services.

Reviewer 2 Report

The authors are commended for their good job in extensively reviewing the literature for CAR T-cell therapy for HCC. CAR T-cell therapy for solid tumors is a challenge that scientists are trying to overcome. The authors did highlight most of the main challenges and possible solutions for CAR T-cell therapy for HCC. However, I believe their manuscript can benefit for some major and minor revisions:

Major revisions (Must be addressed):

- When discussing the challenges related to transport and penetration, the authors need to mention that CAR-T cells may not be able to penetrate tumor tissue through the vascular endothelium due to tumor mechanisms that reduce the secretion of vascular-related factors: ↑endothelin B receptors à ↓ ICAM-1 à prevents T-cell escape from blood vessels.

- When discussing the challenges and solutions for TME, the authors need to mention the contributory effect of high level of extracellular potassium on TME. High K levels can increase the expression of PD-1, suppress the proliferation of central memory and effector memory T cells, inhibit T cell cytokine production, and dampen antitumor cytotoxicity. In fact, modifying the CAR T cell metabolic profile by increasing potassium channels to increase potassium efflux can be helpful.

- The authors did not discuss the challenges and solutions for antigen heterogeneity. It is one of the most important challenges for CAR T-cell therapy in solid malignancies. This is one of the main reasons why CAR T is more effective in hematological malignancies as heme malignant cells homogenously express the same target receptor (for example CD19). On the contrary solid tumors have varying levels of antigen expression and distribution of antigen-positive cells; hence, tumor heterogeneity. There are ways to possibly overcome that such as targeting multiple antigens at the same time. This can be achieved by either co-expressing several CARs on a single T-cell or expressing a chimeric receptor with two or more antigen recognition domains to identify multiple antigens. In fact, there is some research to target cancer stem cells that are closely related to tumor heterogeneity. For example, CD133 is a tumor stem cell marker that is overexpressed in many solid tumors and is now considered a target tumor marker for CAR-T cells.

Minor revisions (strongly recommend addressing):

- I would suggest staring the manuscript with explaining the CAR T receptor targets that can be used for HCC to give the reader an idea on how CAR T can work for HCC.

- Please refrain from using non-scientific terms such as "hot treatment", “battle begins to be fought”.

English language:

- The manuscript needs significant English review and editing.

Author Response

All comments have been responded to one by one in the manuscript. 

Regarding English writing, I have applied for English editing services.

Reviewer 3 Report

In this review, Jie et al. discussed the recent advances and strategies for enhancing the efficacy of CAR-T cells in the treatment of HCC, one of the most common and deadly solid tumors globally. Based on the limitations of CAR-T for HCC including Infiltration obstacles, inhibition CAR-T function from TME, or antigen escape, the authors discussed the corresponding state-of-art updates on CAR-T engineering. The review is comprehensive and relevant to the title aiming to organize the advances in improving the efficacy of CAR-T in the treatment of HCC. However, the English writing and flow of the review may need to be optimized further.

General comments

1. A lot of run-on sentences appeared in the main text in which two or more independent clauses are joined without appropriate punctuation or conjunction. Line 37-39, line 57-58, line 75-78, line 158-160…

2. The descriptions in table 1 should be more precise and compact. Regarding the “Available strategies”, the references should also be shown correspondingly.

3. For table 2, it’s better to separate the references into a new column followed by the column “Current research results in liver cancer”.

4. In part 3, 3.1 and 3.2 have exactly the same subtitle.

5. In part 8, “Other options” seems not well summarized. The authors can illustrate it more carefully by including recent progress/ lessons that can be learned from other fields.

6. Please review your manuscript carefully throughout and revise the grammar, typos…

Author Response

(The authors gave the same response as above.)

Round 2

Reviewer 2 Report

The manuscript looks better now with the edits.

Reviewer 3 Report

The authors did good revisions accordingly.